# Long-Term Periodic and Conditional Survival Trends in Prostate, Testicular, and Penile Cancers in the Nordic Countries, Marking Timing of Improvements

**DOI:** 10.3390/cancers15174261

**Published:** 2023-08-25

**Authors:** Filip Tichanek, Asta Försti, Akseli Hemminki, Otto Hemminki, Kari Hemminki

**Affiliations:** 1Biomedical Center, Faculty of Medicine, Charles University Pilsen, 30605 Pilsen, Czech Republic; f.tichanek@gmail.com; 2Hopp Children’s Cancer Center (KiTZ), 69120 Heidelberg, Germany; a.foersti@dkfz.de; 3Division of Pediatric Neurooncology, German Cancer Research Center (DKFZ), German Cancer Consortium (DKTK), 69120 Heidelberg, Germany; 4Cancer Gene Therapy Group, Translational Immunology Research Program, University of Helsinki, 00014 Helsinki, Finland; akseli.hemminki@helsinki.fi (A.H.); otto.hemminki@helsinki.fi (O.H.); 5Comprehensive Cancer Center, Helsinki University Hospital, 00290 Helsinki, Finland; 6Department of Urology, Helsinki University Hospital, 00029 Helsinki, Finland; 7Division of Cancer Epidemiology, German Cancer Research Center (DKFZ), Im Neuenheimer Feld 580, 69120 Heidelberg, Germany

**Keywords:** time trends, prognosis, periodic survival, conditional survival, treatment

## Abstract

**Simple Summary:**

Male cancers include common prostate cancer (PC) and the much rarer testicular (TC) and penile cancers. Recent survival data for these cancers are relatively good, but long-term studies are rare. To analyzed relative survival in these cancers, we used the NORDCAN database with information from Denmark, Finland, Norway, and Sweden over a 50-year period (1971–2020). Survival improved early for TC, and 5-year survival reached 90% after 1985. Towards the end of the follow-up, TC patients who had survived the 1st year survived the next 4 years with a comparable probability to the background population. For PC, 90% survival was reached after 2000. For penile cancer, 5-year survival never reached 90%, and the improvements in survival were modest at best. As conclusions, more than 90% of the patients diagnosed with PC and TC are alive 5 years later compared to men in general. For penile cancer, mortality is higher, and early symptoms should be discussed with the doctor.

**Abstract:**

Survival studies are important tools for cancer control, but long-term survival data on high-quality cancer registries are lacking for all cancers, including prostate (PC), testicular (TC), and penile cancers. Using generalized additive models and data from the NORDCAN database, we analyzed 1- and 5-year relative survival for these cancers in Denmark (DK), Finland (FI), Norway (NO), and Sweden (SE) over a 50-year period (1971–2020). We additionally estimated conditional 5/1-year survival for patients who survived the 1st year after diagnosis. Survival improved early for TC, and 5-year survival reached 90% between 1985 (SE) and 2000 (FI). Towards the end of the follow-up, the TC patients who had survived the 1st year survived the next 4 years with comparable probability to the background population. For PC, the 90% landmark was reached between 2000 (FI) and after 2010 (DK). For penile cancer, 5-year survival never reached the 90% landmark, and the improvements in survival were modest at best. For TC, early mortality requires attention, whereas late mortality should be tackled for PC. For penile cancer, the relatively high early mortality may suggest delays in diagnosis and would require more public awareness and encouragement of patients to seek medical opinion. In FI, TC and penile cancer patients showed roughly double risk of dying compared to the other Nordic countries, which warrants further study and clinical attention.

## 1. Introduction

Global survival in many types of cancers has developed favorably over the last decades [1,2,3]. From the Nordic countries with their long-term traditions of cancer registration, survival data are available over a half century, confirming the long-term success in cancer control, which, however, varies by cancer type [4]. Although early diagnosis and treatment are considered key determinants of survival, there are numerous factors that directly or indirectly influence survival, including patient care, age, and comorbidities [5]. Interpretation of survival data may be complicated even if changes in incidence can be excluded as a contributing factor [6]. If survival improves shortly after the introduction of a novel therapy, the causal relationship is likely. Examples of undisputed therapeutic gains were the introduction of cisplatin-based therapy for testicular cancer (TC) and imatinib for chronic myeloid leukemia [7,8]. Metastatic cancers confer poor survival, which should be seen as a low 1-year survival. The likelihood of metastatic spread is low in tumors diagnosed early, and thus an alert population, a well-functioning health care system, and improvements in imaging techniques facilitate early detection, which should be seen as improving 1-year survival [4,9]. Even 5-year survival should consequently increase, but 5-year survival alone is not able to point out the time of the improvement without data on 1-year survival [9,10]. Conditional 5/1-year survival describes the survival experience until year 5 in those who survived year 1 and indicates death rates between years 1 and 5 after diagnosis [11]. Another related measure is the difference between 1- and 5-year survival estimates, which is small for cancers of good survival [11,12].

We will assess periodic relative survival in male cancers from Denmark (DK), Finland (FI), Norway (NO), and Sweden (SE) from 1971 to 2020. Risk factors for these cancers are best known for penile cancer, for which human papilloma virus (HPV) infection is the major cause [13]. The origins of TC are thought to lie in the embryonic period, and endocrine disruptive chemicals are assumed to be important risk factors [14]. For prostate cancer (PC), smoking is a risk factor associated with aggressive presentation and poor outcome [15]. In the Nordic countries, patient access to health care is guaranteed with minimal costs, which is an important condition for true population-level survival studies, allowing assessment of “real-world” survival experience. Current survival in these cancers is known to range from excellent in TC and PC (5-year survival >90%) to moderate in penile cancer (70%), but long-tern survival trends are less known [4]. In addition to the standard 1- and 5-year survival, we show data for conditional 5/1-year survival and differences between 1- and 5-year survival. The purpose of the present study is to characterize long-term trends in country-specific survival, estimated as breakpoints in trends and as annual survival changes, which are discussed in terms of the therapeutic and diagnostic landscape and incidence changes [6]. As background to survival analysis, we show concurrent incidence and mortality data for these countries [6].

## 2. Methods

The data were obtained from the NORDCAN database 2.0, originating from the Nordic cancer registries [16,17]. The Nordic cancer registries are population-based with practically complete coverage of cancers and no loss to follow-up until death, end of follow-up, or emigration. The NORDCAN database was accessed at the International Agency for Cancer (IARC)’s website in fall of 2022/winter 2023 (https://nordcan.iarc.fr/en) [18]. Using the NORDCAN tools, we accessed data of incidence, mortality, and 1- and 5-year survival, for which the follow-up was extended until death, emigration, loss of follow-up, or to the end of 2020. Incidence and mortality data were age-standardized for the world standard population. For incidence and mortality data, the starting date was 1961 (the earliest available for all countries). Survival data for relative survival were available from 1971 onwards, and the analysis was based on the cohort survival method for the first nine 5-year periods and a hybrid analysis combining period and cohort survival in the last period 2016–2020 [19,20]. Age-standardized relative survival was estimated using the Pohar Perme estimator [11]. Age-standardization was performed by weighting individual observations using external weights, as defined on the IARC website. Age groups 0 to 89 were considered. The DK, FI, NO, and SE life tables were used to calculate the expected survival. As the age distribution in any cancer differed, age adjustment for each was specifically performed using reference age distribution in each population defined by the International Cancer Survival Standards (ICSSs), with weights for specific age groups (https://nordcan.iarc.fr/en). Incidence and mortality data were obtained from NORDCAN; the stating year 1960 was selected as the first year of data from all countries. It preceded the starting date of survival data but was considered important as unique national incidence data from as early as the 1960s.

Relative 1- and 5-year survival compared survival in cancer patients to the age-adjusted population survival, and 5/1-year survival similarly compared survival between years 1 and 5 after diagnosis. Survival difference between 1- and 5-year relative survival was calculated as 1-year survival % minus 5-year survival %.

For statistical modelling and data visualizations R statistical software (https://www.r-project.org, accessed in winter 2023) was used in the R studio environment (https://posit.co/) [21]. Relative survival trends (NORDCAN 5-year periodic %) were generated using the Gaussian generalized additive models (GAM) with thin plate regression splines in Bayesian framework [21]. Methods for the estimation of the conditional relative survival are described elsewhere [21]. Changes in survival trends were estimated through annual % changes and through “breakpoints”, which marked times when annual changes in survival could be defined with at least 95% plausibility. These are described in the legends to the figures and the detailed estimation methods are available in the above paper [21].

Time trends of 1- and 5-year relative survival (in %; obtained from NORDCAN for each of the 5-year periods) were modelled using the Gaussian generalized additive models (GAM) with thin plate splines (5 knots) and identity links. Models were run in the Bayesian framework using the “brms” R package [22,23], which employed “Stan” software for probabilistic sampling [24]. Separate models were used for different cancers and 1- and 5-year survival. The GAM models included the effect of the *country* and *country*-specific non-linear effect of *time* (timepoint = middle year of each 5 years period) as predictors, allowing estimation of the relative survival across a continuous time scale despite the discrete distribution of data points. As the input data (estimates of the 1- and 5-year survival in each of the 5-year periods) were variably uncertain, standard errors for each data point (obtained from confidence intervals shown in the NORDCAN database) were included in the model. See https://github.com/filip-tichanek/nord_male for commented R code. 

## 3. Results

The total numbers of PC, TC, and penile cancers in the Nordic countries in the 50-year period are shown in Table 1. Considering population sizes, the low number of TC cases in FI and the high number of TC cases in DK and NO is noticeable. The median ages of onset are at around 70 years for PC and penile cancer, 34 years in FI, and 38 years in DK and SE for TC.

Data on incidence and mortality in TC, PC, and penile cancer are presented in Figure 1.

Relative survival in the male cancers in DK is shown in Figure 2. For TC, curves for 1- and 5 years relative survival met each other at around 2005, with the consequence that 5/1-year survival reached close to 100% (Figure 2a). For PC, 1-year survival modestly increased through the follow-up period. The curves for 5- and 5/1-year survival run in parallel, first declined until 1985, turned to a steep increase, which culminated in 2010, and a modest decline followed (Figure 2b). Survival of penile cancer showed a steady increase for all three survival measures (Figure 2c).

In FI, the pattern of survival in TC was similar to the pattern in DK, although the survival in FI was generally lower, and 5-year survival remained below 1-year survival throughout (Figure 3a). Also, survival in PC followed the DK pattern but at a higher level; in FI, there was no initial decline, and the steep increase started 10 years earlier in FI compared to DK (Figure 3b). Survival in penile cancer showed a slight increase, statistically supported only for 5-year survival from 1961 to 1995 (Figure 3c).

In NO, survival patterns for TC and PC resembled those for FI, except that early survival in TC was better in NO than in FI (Figure 4a,b). Survival in penile cancer in NO did not show any clear trend, but all survival plots were higher than in FI (Figure 4c).

In SE, TC survival resembled the DK pattern, and the estimated 1- and 5-year survival curves reached each other before 2010 (Figure 5a). According to Appendix A, 5-year survival was even slightly higher than 1-year survival during the last 10 years. Survival trends for PC were similar to FI and NO (Figure 5b). Survival in penile cancer did not show any clear trend (Figure 5c).

Appendix A points out differences between the countries. Survival in TC was lowest in FI for most of the periods studied: estimated 5-year relative survival in FI exceed 90% around 2000, i.e., 15 years later than in SE (Figure 2, Figure 3, Figure 4 and Figure 5). In the case of PC, there was a rapid increase in 5- and 5/1-year survival during the 1980s/1990s in all countries, but the increase started 10 years later for DK (Figure 2, Figure 3, Figure 4 and Figure 5). In 2016–20, the DK 5-year survival had barely reached 90% compared to 95% for the other countries. In penile cancer, FI showed distinctly lower 5-year (and partially also conditional) survival compared to other Nordic countries: the starting level of 40% for FI was very low, and the final level of below 70% was distinctly below the other countries (Appendix A).

In Appendix A, we show the absolute survival differences between 1- and 5-year survival during the 50 years, during which time a large reduction was observed for TC and PC. For TC, the difference in DK and FI was over 2% units and about 0% units at the end of the follow-up in NO and SE. For PC, the remaining difference was about 5% units. For penile cancer, the difference declined in time for DK and FI but not for NO or SE.

## 4. Discussion

The use of three different survival metrics and their annual changes allows insight into the timing and the underlying factors boosting survival. The three male cancers displayed distinct distributions in these metrics. Survival improved early for TC, and 5-year survival reached 90% already in 1985 in SE, as the first country, and around 2000 in FI as the last country. For PC, reaching the 90% landmark took longer, as it was reached in FI shortly after 2000 but after 2010 in DK. For penile cancer, 5-year survival never reached the 90% landmark, DK came to 85% but FI remained at 70%. While the improvement in survival was overall similar for TC and PC between the Nordic countries, penile cancer survival improved only for DK and FI with low starting levels. We discuss the three cancers individually below.

TC was the first solid cancer for which high cure rates in a metastatic state were reached and which was mainly associated with the application of cisplatin-based chemotherapy. Clinical trials with cisplatin regimens were conducted in the 1970s, and these were adopted as the standard therapy for advanced TC around 1980 [8,25]. We can see from Figure 2, Figure 3, Figure 4 and Figure 5 that 5-year survival in TC developed very well from 1971–75 onwards, and the slope for the 5-year survival curve started to bend down after 1980 (later in FI), probably indicating that the 90%+ cure rates with cisplatin therapy were slowly reached. Additionally, treatment and cure have been individually adapted to the patient’s needs, and the last 5-year survival in the present study was well over 90% in all countries, reaching 98.8% in SE. In the present survival figures, we could see a remarkable time-dependent approaching of the curves for 1- and 5-year survival (in SE, 5-year survival was as high as 1-year survival during the last 10 years) and a 100% concomitant approaching of the curves for 5/1-year conditional survival. The implication is that TC patients had reached the 5-year survival level of the background population; no TC-related extra deaths occurred after year 1. With achieved high survival rates, concerns have arisen about the long-term consequences of successful therapy, as second primary cancers and other medical conditions are recorded in TC survivors [26,27]. The situation is analogous to Hodgkin lymphoma, where well-designed chemotherapy achieved high cure rates, but mortality in second primary cancers called for a revision of the applied chemotherapy regimens [28,29].

For PC in all countries, 1-year survival approached 100%, implying that early deaths were rare. However, 5-year survival improved slower, and, finally, NO and SE reached almost 95% but DK only 90% (DK survival was significantly lower than that for the other countries). The 5/1-year survival closely followed the 5-year survival, indicating that most deaths occurred in the period past year 1. The opportunistic PSA testing, which started around 1990, introduced incidence changes that impeded survival evaluations of a large increase in 5-year survival since 1990 [30,31]. Other studies suggested that PSA testing has led to earlier treatments and decreased PC mortality by around 30% compared to the pre-PSA [32,33]. However, simultaneously, PSA testing has led to significant overtreatment. In DK, the PSA era apparently started later than in the other countries, and, curiously, 5-year survival appeared to decrease before the PSA surge. Another plausible explanation might be national differences in how elevated PSA values were interpreted and when it triggered prostate biopsies. The Nordic countries have national guidelines for diagnostics and treatments of PC, which are adjusted according to the European Society for Medical Oncology (ESMO) and European Association for Urology (EAU) guidelines but, for example, in SE, with some unique recommendations [34,35,36,37]. Generally, risk/stage adapted therapies and diagnostics are recommended. Active surveillance is preferred in less aggressive PC, whereas more aggressive local cancers should typically be treated with prostatectomy or radiotherapy [34,35]. In more advanced states, androgen deprivation therapy and chemotherapy should be applied with the addition of novel agents in castration-resistant cases [34,35].

Penile cancer is a rare cancer, and the treatment recommendations in a metastatic state may not be evidence-based [38]. The relatively poor 1-year survival suggests that diagnosis is derived late in the disease causation. It is known that the prognosis is worsened if more than one inguinal lymph node is affected [38]. Reasons for delayed diagnosis may be prudishness and embarrassment for seeking medical opinion, lack of urological resources, or simply poor knowledge of this rare but easily visible cancer [39]. Accordingly, living alone and in poor socio-economic conditions is linked to poor prognosis [39,40]. Men with unhealing wounds in the sulcus of the penis or growing tumors should seek for immediate medical opinion. Poor development in survival is shared by female HPV-associated cancers of the cervix, vagina, and vulva [41]. For penile cancer, surgical techniques have improved, and radiotherapy and chemotherapy have additionally been used [42]. In accordance with female HPV-related cancers, immunotherapy may be an option in metastatic penile cancer [43]. Nevertheless, the increase in survival is evidenced only for DK according to our results. The current 5-year survival for FI was only 68.6%, NO and SE around 77%, and DK 85.7%, indicating that FI penile cancer patients have double the risk of dying compared to their DK mates.

We tried to find an explanation for the poor survival in penile cancer in FI. Our present resulted showed that there was no large difference between the countries in the median diagnostic ages (FI 69 years, DK and NO 70 years, and SE 71 years). According to the Scandinavian Penile Cancer Group, the major differences in treatment were centralized to only two hospitals in DK and SE, whereas NO and FI maintained a decentralized policy, and FI had as many as 20 surgical departments for penile cancer [44]. However, since 2016, treatment in FI has also been centralized to two hospitals. Disease biology may offer a clue to the poor FI survival (country of lowest incidence) and good DK survival (country of highest incidence). As HPV is the main risk factor of penile cancer, one may assume that DK cases are more often HPV-related than the FI cases. Survival in HPV-associated penile cancers is assumed to be better than in cancers not associated with HPV [45]. This also the case in other HPV-associated cancer, such as oropharyngeal cancer [46].

While we observed differences in the development of survival for the male cancers, the only statistically significant differences for 5-year survival in the final period were for PC, for which DK survival was lower than that in the other countries, and for penile cancer, for which FI survival was lower than that for the best country DK. For many solid cancers, recent developments in survival have been best for DK and NO and worst for FI among the Nordic countries, for reasons that may be related to health care funding and organization [4,35]. Why DK survival in PC was below the other countries requires further investigation; one contributing factor may be in DK treatment guidelines, which recommend antiandrogen monotherapy as primary hormonal therapy for locally advanced, non-metastatic prostate cancer, where curative therapy is not an option [47].

We also observed large incidence differences in the countries, including penile cancer and TC. As for penile cancer, HPV is the main risk factor, and the high incidence in DK and low incidence in FI is probably related to prevalence of HPV infections. Similar differences between DK and FI were for cervical cancer, another HVP-related cancer [41]. The national incidence differences in TC are not well understood, but the DK rates are known to be some of the highest in the world [48].

The limitations in the present study are lacking pathological information of the cancers at diagnosis (particularly relevant for PC) and any treatment information. Another limitation of the NORDCAN data is that it is not possible to carry out age-specific survival analyses. According to literature, diagnostic age is an important determinant of survival in PC, with elderly men being a disadvantaged group [49,50]. For penile cancer, data are sparse, but survival for old patients is worse than for young patients [51]. For TC, survival is better for seminoma than for nonseminoma, both of which are common subtypes, but nonseminoma is an earlier onset disease [52,53]. Histological data are not available in NORDCAN. The advantages of the NORDCAN data are its uniquely long follow-up time from high-level cancer registries. It is not feasible to assume that comparable pathological data were available over 50 years, as it has turned out that even the closely collaborating Nordic cancer registries have difficulties in comparing data on tumor characteristics (stage) for example [54].

## 5. Conclusions

The three male cancers showed different survival histories in the Nordic countries. Survival rates increased constantly for TC and could reach a population level of survival after year 1 of diagnosis. For PC after 2000, mortality by year 1 was nil, but late mortality requires attention. For penile cancer, no or small survival improvements could be observed over the 50 years, whereas data from DK indicate that progress could be made. The relatively high early mortality may suggest delays in diagnosis, which may be due to medical and social factors in a “neglected” cancer. In FI, TC and penile cancer patients had a 2-fold risk of dying compared to their Nordic mates, which warrants clinical attention.

## Figures and Tables

**Figure 1 cancers-15-04261-f001:**
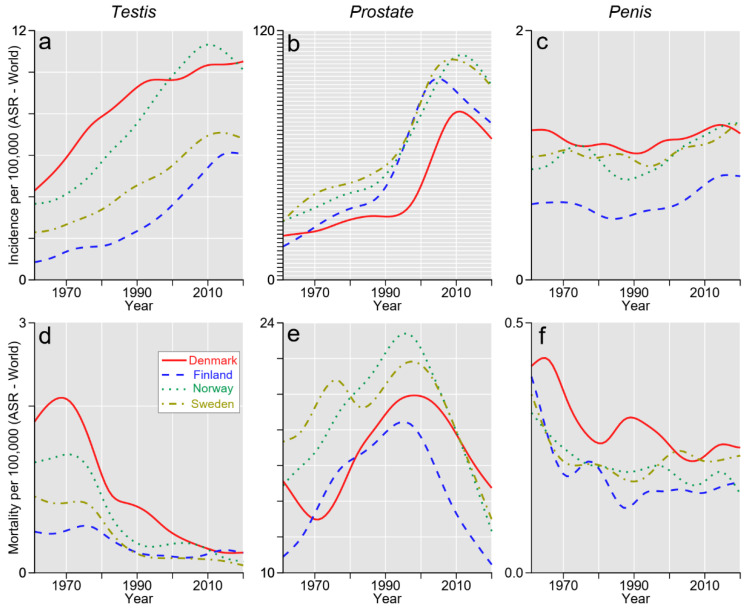
Incidence (**a**–**c**) and mortality (**d**–**f**) in male-associated cancers of following localizations: (**a**,**d**) testis, (**b**,**e**) prostate, and (**c**,**f**) penis. The figure was created in R using data from Nordcan. Lines were smoothed via cubic smoothing spline.

**Figure 2 cancers-15-04261-f002:**
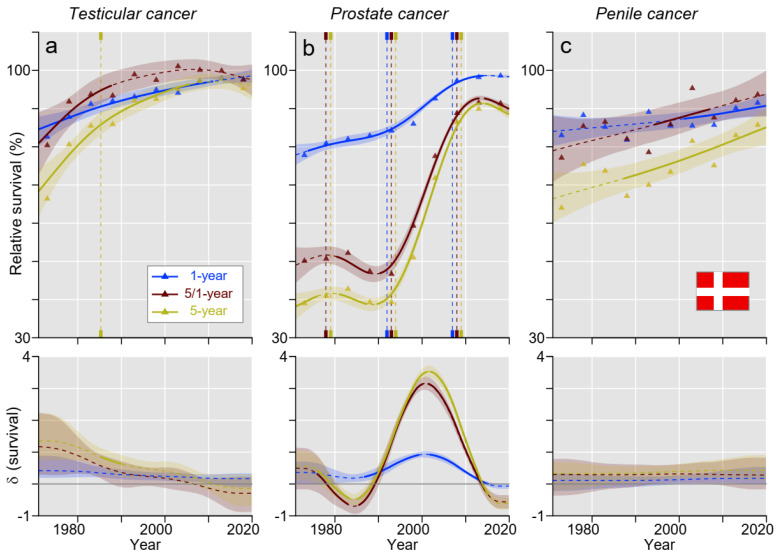
Relative 1-, 5/1- (conditional), and 5-year survival in Danish men with (**a**) testicular, (**b**) prostate, and (**c**) penile cancer. The vertical lines mark a significant change in the survival trends (“breaking points”), and the bottom curves show estimated annual changes in survival. The curves are solid if there is >95% plausibility of the growth or decline. Shadow areas indicate 95% credible intervals. All curves are color coded (see the insert).

**Figure 3 cancers-15-04261-f003:**
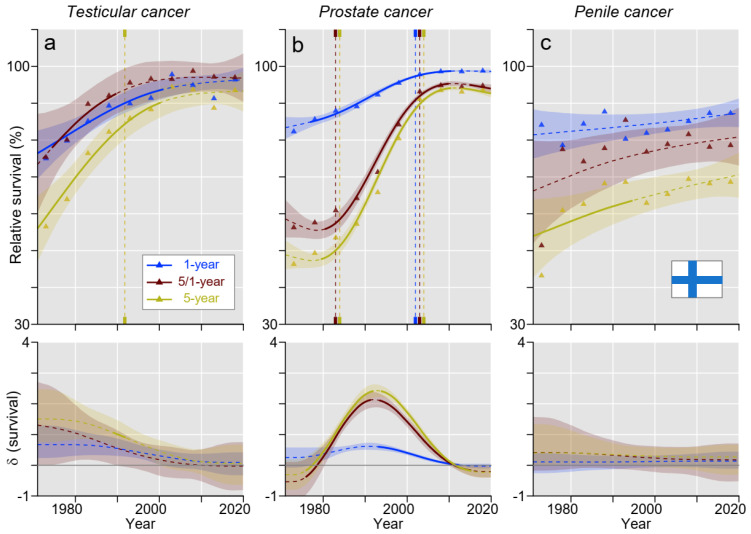
Relative 1-, 5/1- (conditional), and 5-year survival in Finnish men with (**a**) testicular, (**b**) prostate, and (**c**) penile cancer. The vertical lines mark a significant change in the survival trends (“breaking points”), and the bottom curves show estimated annual changes in survival. The curves are solid if there is >95% plausibility of the growth or decline. Shadow areas indicate 95% credible intervals. All curves are color coded (see the insert).

**Figure 4 cancers-15-04261-f004:**
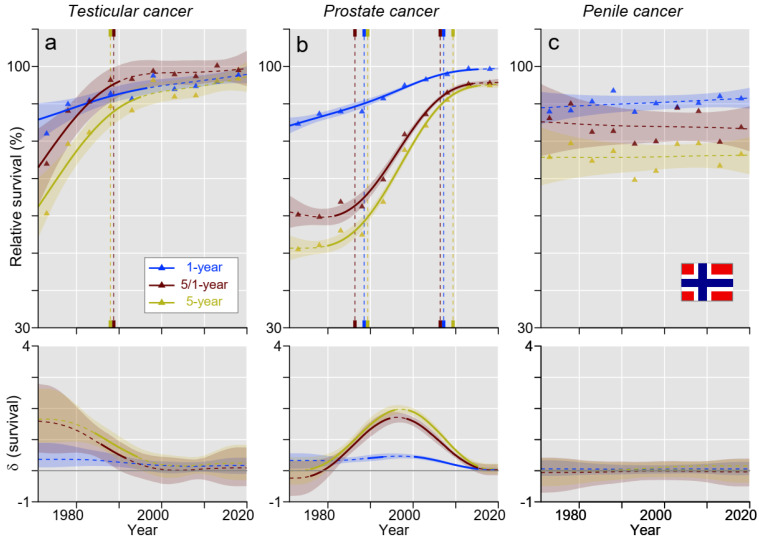
Relative 1-, 5/1- (conditional), and 5-year survival in Norwegian men with (**a**) testicular, (**b**) prostate, and (**c**) penile cancer. The vertical lines mark a significant change in the survival trends (“breaking points”), and the bottom curves show estimated annual changes in survival. The curves are solid if there is >95% plausibility of the growth or decline. Shadow areas indicate 95% credible intervals. All curves are color coded (see the insert).

**Figure 5 cancers-15-04261-f005:**
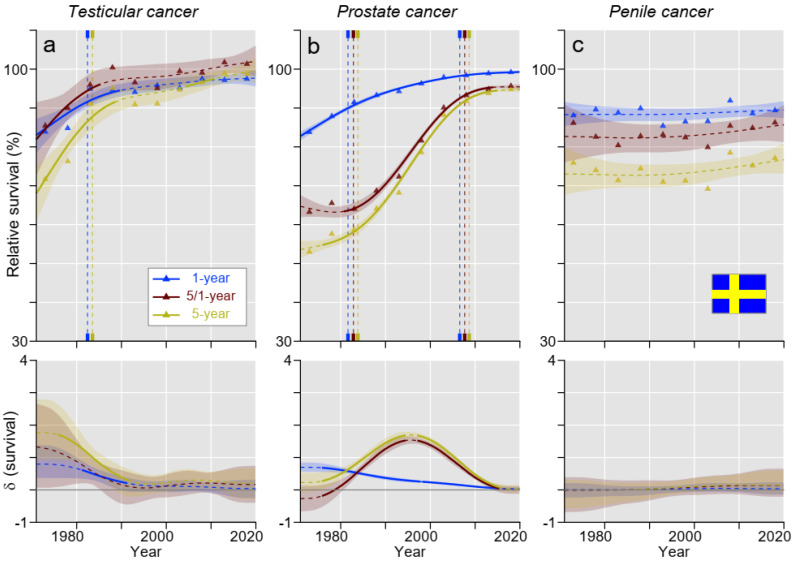
Relative 1-, 5/1- (conditional), and 5-year survival in Swedish men with (**a**) testicular, (**b**) prostate, and (**c**) penile cancer. The vertical lines mark a significant change in the survival trends (“breaking points”), and the bottom curves show estimated annual changes in survival. The curves are solid if there is >95% plausibility of the growth or decline. Shadow areas indicate 95% credible intervals. All curves are color coded (see the insert).

**Table 1 cancers-15-04261-t001:** Case numbers of male cancers in the Nordic countries 1971–2020 and their estimated median ages at onset in 2011–2020.

Population	Prostate		Testis		Penis	
	N	Age	N	Age	N	Age
Denmark	122,150	69	13,141	38	2500	70
Finland	140,571	71	4507	34	1197	69
Norway	145,648	69	10,278	36	1913	70
Sweden	336,314	70	11,966	38	4196	71

## Data Availability

Aggregated data are publicly accessible at the IARC web site (https://nordcan.iarc.fr/en, accessed in 2022/2023).

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
