# Peer review of "Long-Term Periodic and Conditional Survival Trends in Prostate, Testicular, and Penile Cancers in the Nordic Countries, Marking Timing of Improvements"

_cancers, 2023, doi:10.3390/cancers15174261_

Round 1

Reviewer 1 Report (New Reviewer)

From a biostats and clinical epidemiology point of view, this registry study has been very well planned and reported. I have no major concerns, nevertheless I'd like to underline that PC treatment has deeply evolved on time course (chemotherapy, radiotherapy, open surgery, laparoscopic and robotic surgery, immunotherapy, old and new antiandrogen molecules) much more than for TC and penile cancer. This topic should be more clearly stated.

All supplements are unavailable in the website

Author Response

Thank you for the comments. PC treatment was described in more detail in l. 260-8 with several references. 

Reviewer 2 Report (New Reviewer)

This is an interesting and well-written manuscript. Only minor revisions are needed before it can be published:

1)   I suggest providing references for lines 58 to 63 in the introduction section.

2)   Please beware of spelling mistakes, such as “ long tern” in line 78 and “Finish” in Figure 2.

3)   The colour scheme of the figures does not allow the curves to be distinguished when printed in black and white. You should ensure figures remain meaningful even if converted to grayscale. The best option is to distinguish not only by colour but also by different line styles (dashed, dotted etc..) and varying colour brightness.

Author Response

Thank you for the comments. References were added to l. 62, 64. Finnish was corrected and all figures were revised to allow black and white viewing.

Reviewer 3 Report (New Reviewer)

In the manuscript "LONG-TERM PERIODIC AND CONDITIONAL SURVIVAL TRENDS IN PROSTATE, TESTICULAR AND PENILE CANCERS IN THE NORDIC COUNTRIES MARKING TIMING OF IMPROVEMENTS" the Authors illustrate the survival statistics for male genital tract tumours in a long period of observation in the cancer registries of four Northern European countries. The long-term prognosis of these tumours has been good for many years; the relative survivals at 1 year, at 5 years, and at 5 years in the survivors after the first year have already reached high values in the early 2000s. The observed trends are been influenced by the incidence of individual tumours, their risk factors, and diagnostic interventions (the PSA test in prostate cancer) which have modified the approach to the disease.

The authors accurately illustrate the techniques of statistical analysis of the data collected from the Northern European cancer registries, allowing the reader to understand the methodology used, as well as identify in detail the recognized risk factors for the various cancers. The authors should help readers more easily access the incidence data they have used in compiling survival statistics. In the Methods section, in the presentation of statistical methods, the authors refer to their recent publications (see ref. 9) or in any case to works in which the procedures they used have been applied (ref. 19) rather than indicating the works reference methods.

Very explanatory graphs accompany the exposition of the results, however I cannot find the supplementary material mentioned in the text with the comparisons between the different countries. In this regard, I reiterate the importance of providing rapid access to data on the incidence trends for the pathologies described, also with details of the specific rates by age, given the different ages of incidence of these tumours.

In the part of the discussion on prostate cancer it would be useful to have more details on the diffusion and availability of the PSA test over time and in the different countries observed and in general to be able to evaluate the risk factors in a comparative way.

Overall, I found this manuscript interesting, however with the suggested additions it would acquire a greater depth.

Author Response

Thank you for you comments. Incidence and mortality data are now presented as Fig. 1. Median diagnostic ages were added to Table 1. PSA screening initiation dates were added to l. 252, 254 with references.

Round 2

Reviewer 3 Report (New Reviewer)

I have previously suggested emphasizing the differences in the age of incidence of the three tumours considered, but I don't see any hints made to that effect. The ability to process data from cancer registries must make it possible to highlight the richness of this heritage, especially when it is done with a scientific publication, without taking for granted the information connected (incidence) to the topic dealt with (the survival). I find it very useful that direct access to the age-specific rates of these cancers in the supplementary data is facilitated and that they are mentioned in the manuscript

Author Response

We had some difficult in understand the point in the age related comments. We had added median age of onset to Table 1, showing the much earlier onset for TC than for the other tumors. In relative survival analysis each cancer is separately age-adjusted to the population age structure so each cancer is analyzed uniquely. We added this to methods, l. 103-6.

NORDCAN has received defined types of data set from the Nordic cancer registries (explained in l. 87). However age-specific survival analysis is not possible in NORDCAN. Thus we added this to the limitations but also added literature data about age-related survival in the present cancers with related references . l. 217-24.   

This manuscript is a resubmission of an earlier submission. The following is a list of the peer review reports and author responses from that submission.

Round 1

Reviewer 1 Report

This study provides valuable insights into the long-term survival rates of prostate, testicular, and penile cancers in Denmark, Finland, Norway, and Sweden over a 50-year period. The use of a robust database like NORDCAN and the application of generalized additive models lend credibility to the findings. However, several aspects warrant improvement and further consideration.

Major comments:

1-     [Introduction] The structure and contents of the Introduction section need to be improved throughout. There seems to be a lack of connection between the content of each paragraph and within each paragraph. The second paragraph, in particular, appears to have a weak relation to the purpose of this study. The background and purpose of the study are not clearly stated, and there is insufficient information support to demonstrate the necessity and importance of the study. I recommend addressing these issues to enhance the clarity and coherence of the Introduction.

2-     [Introduction] In connection to the previous point, do the lines 53-57 have relevance to the main idea of this study?

3-     [Methods] It is suggested that the Methods section should include information about the 5-year/1-year survival and the 1- and 5-year relative survival differences, as these results are presented in figures and supplementary tables.

4-     [Results] Could you please provide more detailed information about the database? For example, it would be helpful to know the number of cancer registrations used in the analysis for each country, as well as the average or median follow-up period. The lack of this fundamental information may impede readers from comprehending the data. Would it be possible to add a table that summarizes the characteristics of the target population?

5-     [Discussion] While the study provides an overview of survival rates across different countries, it does not delve into the potential reasons behind the observed differences. Factors such as healthcare access, screening programs, and treatment protocols could significantly influence survival rates and should be explored.

6-     [Discussion] The study points out the high early mortality in penile cancer, suggesting delays in diagnosis. However, it stops short of providing concrete recommendations for improving public awareness and encouraging patients to seek medical opinion.

7-     [Discussion] The finding that penile cancer patients in Finland have roughly double the risk of dying compared to Denmark is concerning and warrants further investigation. It is important to consider various factors that could impact survival rates, such as medical resources, screening and treatment methods for penile cancer, and the age and health status of patients. In discussing this finding, it is important to determine if appropriate statistical tests were conducted and if potential confounding factors were taken into account. For instance, it is possible that patients in Finland are older or have poorer health conditions, which could increase their risk of death, but this may not necessarily be due to regional differences. It would be beneficial to analyze the backgrounds of cancer patients to gain a better understanding of the situation.

Minor comments:

1-     Both "5-year" and "5-years" are being used in the manuscript, but I think "5-year" is the correct term. I suggest that you consistently use "5-year" throughout the manuscript for clarity and consistency.

2-     "5y/1y survival", "5-year/1-year survival" and "5/1 survival " are being used in the manuscript. I suggest that you use consistent term in all relevant contexts throughout the manuscript for clarity and consistency.

3-     Would it be better to provide specific numerical values in lines 77-79?

4-     Could you verify if there was a rapid increase in the number of prostate cancer during the 1990s rather the 1980s in lines 150-152? Besides, could it be 5~10 instead of ~10? Please verify.

5-     [Supplementary Table 2] Could you confirm that the 5-year/1-year survival and the 4-year conditional survival are interchangeable terms that express the same meaning in survival analysis. I think they are two distinct measures.

6-     Please review the description in lines 158-159 as it appears to be inconsistent with the corresponding table.

7-     Could it be 35% instead of 35 in line 160? Please confirm.

Reviewer 2 Report

I found this study and manuscript difficult to follow as written.  Though underlying premise for analysis is of interest, results are unclear and conclusions are speculative.

This manuscript requires extensive editting

Reviewer 3 Report

This paper examines the long-term periodic and conditional survival trends in the prostate, testicular and penile cancers in the Nordic countries. The authors have analyzed 1-year and 5-year relative and 5/1 conditional survival for three cancers using the NORDCAN database with information from Denmark, Finland, Norway, and Sweden over 50 years (1971-2020). Here are my comments

The study shows overall survival rates of 1, 5 relative survival, and additionally conditional 5/1-year survival. Except for PC, the other two cancer show a very high survival rate. The study would be interesting and add to the literature if the predictors were evaluated for increased relative survival. It is well known that stage, age, comorbidities, and social determinants of health impact survival. The study would then benefit from knowing the factors that are important to achieving>90% of relative survival. I don’t find the study very impactful except that the authors have added conditional survival.

The incidence of cancer in the four countries varies as seen in the supplementary Fig 1, why do think there is a difference in the incidence? Why some countries have achieved better survival rates comparing the four countries? I think the discussion needs more in-depth information.

Minor comment:

Supplementary Fig 1 shows Incidence (a-c) and mortality (d-f) in male-associated cancers of testis, prostate, and penis cancers from 1970 to 2010. It is not clear in the methods why this period was considered.